# Americans experience a false social reality by underestimating popular climate policy support by nearly half

Gregg Sparkman[1] ✉, Nathan Geiger[2] & Elke U. Weber [3]

Pluralistic ignorance—a shared misperception of how others think or behave—poses a challenge to collective action on problems like climate change. Using a representative sample of Americans ($N = 6119$), we examine whether Americans accurately perceive national concern about climate change and support for mitigating policies. We find a form of pluralistic ignorance that we describe as a *false social reality*: a near universal perception of public opinion that is the opposite of true public sentiment. Specifically, 80–90% of Americans underestimate the prevalence of support for major climate change mitigation policies and climate concern. While 66–80% Americans support these policies, Americans estimate the prevalence to only be between 37–43% on average. Thus, supporters of climate policies outnumber opponents two to one, while Americans falsely perceive nearly the opposite to be true. Further, Americans in every state and every assessed demographic underestimate support across all polices tested. Preliminary evidence suggests three sources of these misperceptions: (i) consistent with a false consensus effect, respondents who support these policies less (conservatives) underestimate support by a greater degree; controlling for one's own personal politics, (ii) exposure to more conservative local norms and (iii) consuming conservative news correspond to greater misperceptions.

Addressing a collective action problem like climate change requires individuals to recognize the problem as a threat and to engage in coordinated actions that result in major structural and social change. Collective action problems pose a difficult challenge as individuals are less likely to act when there are others who standby and do nothing—and this outcome is only more common when the problem at hand is not clearly perceived to be a threat[1]. Further, research on threat perception suggests we rely heavily on others' reactions to recognize complex or non-immediate problems like climate change as a threat[2,3]. Similarly, perceptions of social norms, including perceiving others' concern and desire for action, may be key to coordinating collective solutions[4,5], including in environmental domains[6,7]. But such processes are hampered if people fail to accurately perceive that others are concerned and support taking action. Systematic misperception of public opinion (i.e., "pluralistic ignorance"[8]) like a widespread underestimation of public support for climate action could inhibit willingness to talk about the problem with others[8,9], and could lead people to falsely conclude that the vocal minority who dismiss climate change are representative of broader public opinion[10]. Further, given that most Americans report concern about climate change and support many policies to address the issue[11,12], why has the US not yet enacted major climate policy to address the issue? If most Americans were unaware of the popularity of their pro-climate action views, this could encourage inaction through pressures to conform to the (mis)perceived political attitudes of others, a phenomenon robust across the political spectrum[13]. These concerning possibilities raise the question:

[1]Department of Psychology and Neuroscience, Boston College, Chestnut Hill, MA 02467, USA. [2]Media School, Indiana University Bloomington, Bloomington, IN 47405, USA. [3]Andlinger Center for Energy and Environment, Princeton University, Princeton, NJ 08540, USA. ✉e-mail: gregg.sparkman@bc.edu

Do Americans accurately perceive public support for climate mitigation?

Generally, our perceptions about the world, including the social world, are shaped by society and can be thought of as part of a "social reality", wherein some social truths are widely held and can exert influence over us[14,15]. Notably, whether or not these perceptions are accurate, they can shape our actions and beliefs, including our expectations or judgment of others[16]. To better understand the impact social realities, the role of second order beliefs (our beliefs about others' beliefs) are increasingly highlighted as important contributors to and intervention point for contemporary social problems[17,18]. Indeed, there have been calls for a better understanding of the social determinants of collective behavior to be elevated to a major "crisis discipline" of our time[19]. Here, we investigate norm misperception in the climate policy context. Pluralistic ignorance refers to a systematic and shared misperception of a norm, where many people have the same misconception about what most people do or think[20]. For instance, college students have been found to collectively misperceive that drinking is more common and desirable among their fellow students than is actually the case[21]. In addition to perceptions of local community norms, pluralistic ignorance can also pertain to society-wide misperceptions[17]. For example, people may succumb to the "conservative bias", whereby perceptions of public opinion lag behind actual public opinion by some decades, failing to reflect changes and anchoring on historic levels[22]. The conservative bias is particularly likely when public opinion has recently changed on a topic but policy and structural change has not yet resulted from this shift, leaving little concrete indication of a shift in norms[22].

Previous work suggests there may be pluralistic ignorance on climate concern and some related policies amongst the US public. First, research on a related topic suggests that people systematically overestimate the percentage of others in their country who outright reject the existence of human-caused climate change, with representative samples from US, China, and Australia showing that although most in each country believe in man-made climate change, people underestimate the extent to which their fellow citizens do[23–25]. And, research using student samples has found that college students underestimate their peers concern about climate change, broadly[8]. More pertinent to climate policy, one study found that an online convenience sample underestimated Americans' support to regulate CO2 as a pollutant (broadly, not in a specific policy framework), and concern about climate change[26]. Similarly, research on a sample of U.S. congressional staffers found that many underestimated the popularity of carbon pollution restrictions among the public in their district[27]. Further, work in the northeastern U.S. coastal states using an online convenience sample of Americans found that most underestimate support for regional decarbonization approaches like expanding offshore wind[28]. While piecemeal and unrepresentative, these scattered indicators are a cause for concern—one that demands a conclusive investigation of climate policy pluralistic ignorance using a representative sample to investigate concrete, major national climate policies.

Given the possible role of pluralistic ignorance in stalling progress on this existential threat, it is prudent to investigate fundamental, unanswered questions about pluralistic ignorance on climate policy support: Is pluralistic ignorance around climate policy common in the U.S.? Is it contained to specific pockets of Americans, or does it span many populations over many geographies? Does it affect only specific policies, or does it hold for a variety of climate change mitigation policies (e.g., those utilizing market instruments, as opposed to mandates, or direct investment in infrastructure)? As prior research has found that Americans underestimate fellow Americans' belief in climate change[23], it's possible that they also underestimate public support across a range of climate change mitigation policies (i.e., they may expect lower policy support for any policy addressing a problem if

they assume others don't believe that the problem exists). If pluralistic ignorance is present in this context, how large are the misperceptions? Do some misperceptions exist, but perceptions are accurate about majority and minority opinion? Or do they surpass this level and result in misperceiving what the majority of Americans support?

The main objective of the present study is to provide clear and granular answers to the above questions. Additionally, there may be questions about the possible sources of misperception. One contributor to misperceiving popular opinion could be false consensus, where people pay selective attention to others' beliefs that are similar to their own, and overestimate the number of people who agree with them[29]. Indeed, false consensus effects exist for belief in climate change and support for local renewable energy projects in the Northeast US[24,28]. As such, we might expect that U.S. conservatives underestimate support for climate mitigation policies to a greater extent as they have lower approval of said policies.

Second, when forming estimates of frequency or probability, people's guesses are generally shaped by information that is more easily available or retrievable, reflecting an availability heuristic[30]. Thus, people's estimates of national public opinion may show an outsized influence of local norms that are easier to witness firsthand or recall. Given this, people's estimates of the nation as a whole may be swayed by their state-level norms such that those in more conservative states and those in states with fewer climate protests may underestimate climate policy support to a greater degree.

A third possible contributor is media consumption, particularly if media misrepresent public opinion[31]. News media coverage of scientific experts in the U.S. has historically given disproportionately too much time to climate change deniers[32] and presentations of conservatives as oppositional to climate change policy, while the conservative electorate is actually fairly divided on these issues[12]. Given differences in media coverage, one might expect that public opinion misperceptions would be particularly pronounced amongst those who consume news outlets that have been shown to be more inaccuracy-prone[33].

In the present work, we investigate national misperceptions of support for transformative climate policies and broader concern about climate change and show that Americans experience pluralistic ignorance to such a magnitude and breadth that it can be considered a *false social reality*: Americans from all walks of life systematically underestimate public concern about climate change and policy support over a range of climate policies. The magnitude of the effect is such that those who want action are a supermajority (i.e., 66% or higher), while there is a ubiquitous perception across demographics that they are only a minority.

## Results
### Study overview

To create a detailed picture of the state of pluralistic ignorance for climate policy, we use a large stratified sample of US adults ($N = 6,119$) through the Ipsos eNation Omnibus nationally representative panel to compare public opinion on climate change to perceptions of popularity of those same opinions. We commissioned this panel to oversample less-populous states to assess the extent of pluralistic ignorance for each state with greater precision, and aiming for a 10% margin of error for all states. For the full sample, this sample size is more than 80% powered to detect small national levels of pluralistic ignorance (effects as small as $d = 0.04$), as well as being 80% powered to detect separate levels among Democrat, Republican, and Independent partisan groups (effects as small as $d = 0.07$), allowing for very granular comparisons. For all national-level analyses, we applied weights from the survey provider to ensure representativeness (e.g., down-weighting data from smaller states that we oversampled).

Actual levels of U.S. public support on climate policies were obtained from nationally representative public opinion data available

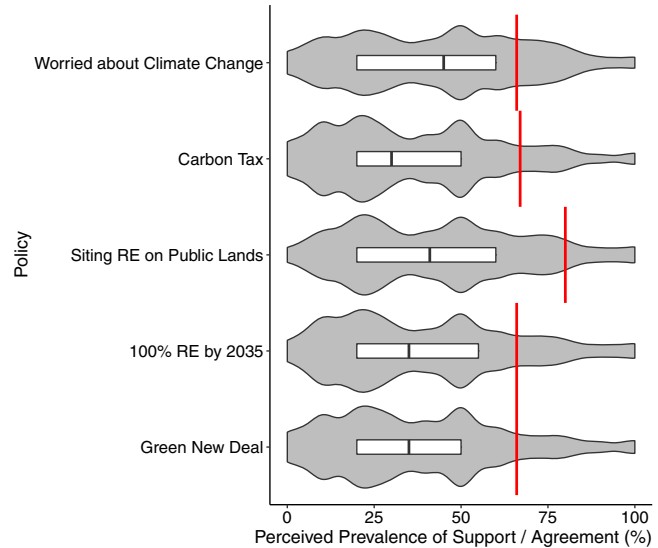

**Fig. 1 | Perceived climate change worry and support for climate policies compared to actual levels.** Boxes inside the violin plot represent the middle 50% of the sample, with a line at the median, while the minima and maxima illustrated represent the full range of responses (from 0–100%). The red line represents the true prevalence, as indicated by contemporaneous national polling[12,43]. RE refers to renewable energy. $N = 6119$ survey participants.

**Table 1 | Differences in real vs perceived national policy support**

| Policy | Actual support (%) | Perceived support (%) | T | Cohen's d | 95% CI of diff. (%) |
|---|---|---|---|---|---|
| Carbon Tax | 67 | 36.6 | 96.26 | 1.27 | 29.7–31.0 |
| Siting RE | 80 | 43.4 | 113.10 | 1.48 | 35.9–37.2 |
| 100% RE | 66 | 39.5 | 69.79 | 0.91 | 21.9–23.2 |
| GND | 66 | 37.9 | 89.81 | 1.17 | 27.5–28.7 |

*Note.* Real, perceived and 95% CI are all in percentages. One sample *t*-tests (two-tailed) were conducted against constant for real policy support values and each have 6118 degrees of freedom. RE refers to renewable energy. GND refers to the Green New Deal.

environmental legislation, as research shows that bundling more redistributive, social equity, and job-creating measures into major environmental policies makes them more popular[35].

Finally, we asked participants about their news consumption, political affiliation, and demographic characteristics, which we used to conduct an exploratory cross-sectional analysis of possible sources of pluralistic ignorance in norm perceptions.

## Prevalence and magnitude of pluralistic ignorance

Figure 1 shows that the vast majority of Americans greatly underestimate how many of their fellow Americans worry about climate change and support transformative climate policies to remedy the situation. While most Americans believe that less than half of the country is worried about climate change ($M_{est} = 43.3$), in actuality it is two-thirds ($M_{real} = 66$), $t(6118) = 70.9$, $d = 0.92$, $P < 0.001$, 95% $CI_{diff} =$ [22.0, 23.3] (see Methods for additional notes on these analyses). Americans' estimates for major climate policy support is the same or even lower ($M$s = 37–43%), when in fact two-thirds of the country or more support each of these policies (Table 1). The distributions of these estimates in Fig. 1 show two noticeable concentrations, one at around 50% and another around 25%, salient proportions that seem to serve as focal points for answering these questions, even though a similarly salient proportion—75%—would have been a far more accurate answer. The misperceptions in estimates are so robust that, for every item assessed, the estimates of the lowest 25% and of the middle 50% of respondents falls well below the true values. More precisely, between 79% and 88% of our national sample underestimate public concern or each policy support.

We also asked participants for estimates of support in their home state, and found these perceptions (when averaged across states) to not vary substantially from the national-level estimates and to have very similar distributions (see Supplementary Fig. 1 in the Supplementary Information File). Overall, this pattern of results suggests that people misperceive support for climate action broadly, having non-specific and robust misestimates for support for a variety of climate policies. Indeed, in an exploratory factor analysis of the five responses shown in Fig. 1, a single factor emerged (all other factors had eigenvalues <1).

## Pluralistic ignorance across partisans and policies

Breaking these perceptions of national public concern and policy support down by partisan politics, we found that Democrats, Independents, and Republicans all estimate levels for climate concern and climate policy support below 50%, while actual values are much higher. However, Fig. 2 shows that Republicans' estimates were reliably lower than Democrats' by 5–12%, with Independents falling somewhere in between.

For the national policy items, contemporaneous polling was available broken down by partisans, so we can compare partisans' estimates of nationwide support to actual partisan levels of support. Figure 2 shows that when Democrats, Independents, and Republicans

from Yale's Program on Climate Change Communication (YPCCC), who used the same survey provider who collected the data for the present study (which may help account for any surveyor-specific sampling or data practices)[11,12]. We also use YPCCC's state level estimates of public opinion (which are estimated from their national survey data via multilevel regression with post-stratification). These data were collected during the same year as the data in the present study (see Methods). Further, to avoid any differences in policy support estimates due to item wording in comparing actual levels to responses from our panel, we used precisely the same policy descriptions as used by YPCCC. Thus, if wording for any of the policy items is subtly leading to inflated (or deflated) support numbers, this wording should also lead to inflated (or deflated) estimates of opinion estimates; thus, specific item wording would be unlikely to create discrepancies between actual and perceived policy support.

We asked participants to estimate the percent of Americans who were at least somewhat concerned about climate change (see Methods for full survey text, and a discussion about using the phrase "climate change" vs "global warming" for this item). We then chose a set of specific climate policies especially relevant to the decarbonization of the US and the attainment of climate mitigation goals such as the 2015 Paris Agreement. We intentionally selected a set of climate change mitigation policies that varied in core features such as utilizing market instruments as opposed to mandates, or those that facilitate investment and the creation of infrastructure. For each policy, we asked participants to estimate the percent of Americans who would support it. Our list of policies included support for a carbon tax levied against fossil fuel companies and redistributed to Americans through tax breaks. The list also included a renewable energy standard that mandates 100% electricity generated by renewable energy in the near term—an essential step in decarbonizing our energy production[34]. And, as decarbonizing our energy infrastructure will require rapidly siting of wind and solar across the US, we also included support for siting renewables on public lands[34]. Given the need to consider infrastructure, jobs, and social equity in transitioning to renewable energy, we also asked participants to estimate the support for the Green New Deal (GND). Notably, large environmental policy packages like the GND and the American Jobs Plan may play a key role in passing

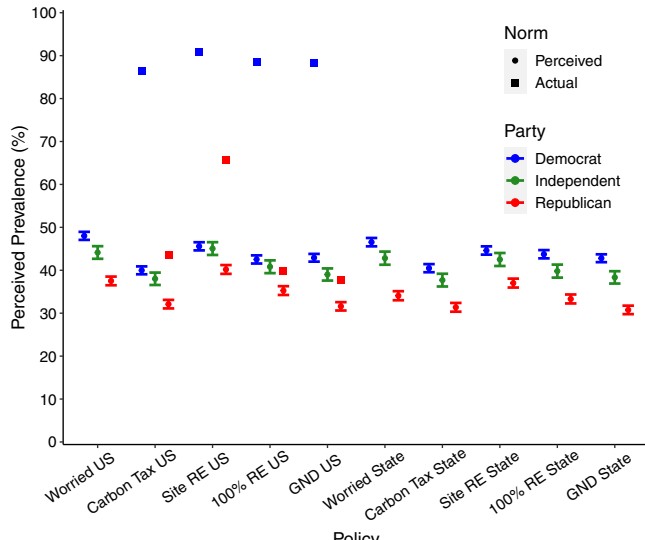

**Fig. 2 | Partisan perceptions of popular climate worry and support for climate policy at national and state levels.** All judgments were elicited for everyone in either the entire nation or one's state (not only for one's fellow partisans). Items on the left labeled with "US" indicate estimates of support for policies at the national level, while those on the right labeled with "State" indicate estimates of support for those in participants' own state. Actual partisan support levels available from contemporaneous polling (available for four US policies[12]) are indicated by squares. Error bars represent 95% confidence intervals for the norm perception means illustrated. RE refers to renewable energy. GND refers to the Green New Deal. $N = 6119$ survey participants, including 2777 Democrats, 1022 Independents, and 2320 Republicans.

estimate how the nation feels on these issues, their estimates of other Americans' support for these policies only really resemble actual Republican levels of policy support. In fact, even if individuals' estimates for the nation as a whole were, for some reason, based solely on Republican levels of support, all partisan groups would still be underestimating support for policies like a carbon tax and siting renewables on public lands. While differences between partisans are consistent with false consensus effects (e.g., Democrats—who are more likely to personally support climate policy—tend to provide relatively higher estimates of others' policy support than do Republicans), these effects are dwarfed by the absolute levels of misperception held by all Americans that strongly underestimates climate policy support.

For all policies and concern for climate change at the national level, as well as state level support for a carbon tax and concern for climate change, we were able to compare each participant's norm estimates to available contemporaneous data on actual support levels. Doing so, we create difference scores for each participant's estimate, which can be aggregated into pluralistic ignorance levels that we analyze below.

Reflecting the lower norm estimates by Republicans, Fig. 3 shows that Republicans' opinion misperceptions are stronger in magnitude than Democrats' and Independents' across all items. Further, we find that all partisan groups underestimate concern for climate change at both the national and state level by roughly 20–30%. In policy support, we find that the magnitude of misperception is highest for support to site renewables on public lands, with underestimates closer to 35–40%. Underestimation is smaller for support for 100-percent renewable energy mandates, which is still between 20–25% lower than actual levels. Support for a carbon tax and a Green New Deal fall in between these levels.

We can also directly compare state and national pluralistic ignorance levels for the two items for which we have data for both (a carbon tax and worry about climate change) to test if estimates are

more accurate for state than national items. Using a mixed model to predict pluralistic ignorance levels across these four items using a dummy-coded fixed effect for item location (state = 0 vs nation = 1) and random intercepts for participant and item type (carbon tax vs worry), we find no difference between the two, $t(21762) = -0.94$, $P = 0.350$, suggesting that people are equally inaccurate at estimating opinions of fellow denizens of their own state, relative to the entire US public.

### Regional variation
Aggregating levels of pluralistic ignorance by state, we can map the magnitude of false norm perceptions across the country. Figure 4 shows that residents of *all* states underestimate how much the nation is worried about climate change and support climate policy (for separate maps for each policy, see Supplemental Fig. 2 and Supplemental Fig. 3). For both perceived popular worry and climate policy support, we see that the southern gulf states (e.g., Mississippi) tend to show the highest pluralistic ignorance. But, underscoring the ubiquity of this misperception, even liberal states such as California and New York underestimate climate policy support as much as many conservative states. In fact, no state was less than 20% off in their estimates of climate policy support. These errors are robust for the more proximal state-level estimates as well, where participants of virtually every state underestimated how concerned their fellow state residents were about climate change, and how much they supported a carbon tax (see Supplemental Fig. 4).

### Variation by demographics
We regressed pluralistic ignorance across items in a linear mixed model weighted to be nationally representative, with random intercept for participant and item, on the full battery of demographics assessed (see Methods). As shown in Supplementary Table 2, we find a number of statistically significant factors. Consistent with false consensus effects, we find participant's political orientation has a notable effect (with 22% underestimation for those who are very liberal to 33% for those who are very conservative). Race also has a notable impact, with 25% underestimation for white respondents to 35% underestimation for black respondents, and other races falling in between. Other demographic characteristics had smaller, but still statistically significant effects. For instance, those living in urban areas were about 29% off, while their suburban counterparts were 26% off (and rural respondents falling in between).

Notably, there was no demographic group for which the estimated range reached accurate levels—instead all groups assessed were at least 20% off. Further, some demographics which might have been anticipated to predict reduced misperceptions did have statistically significant effects, but were small shifts in absolute terms: Those who attended 12 years of schooling but never obtained a GED or diploma were 28% off, while those with a doctorate were still 27% off, just a single percentage point better.

### Variation by local norms
In exploratory analyses, we assessed two state level predictors for their relationship to pluralistic ignorance levels across items with both known real and perceived levels: the voting margin for Biden in the 2020 election (used as a state-level proxy for prevalent political ideology), and the logged number of climate or environmental protests per capita (see Methods). The effects of these predictors were assessed in a multiple regression mixed model with random intercepts for participant and item, and controlling for the top five demographic variables shown to have an effect and likely vary by state (personal political orientation, race, employment status, age, and income). Consistent with an availability heuristic, we find that both indicators of local norms influence norm estimates: there is a significant effect for state political ideology $b = -0.02$, $t(39760) = 2.35$, $P = 0.019$, such that

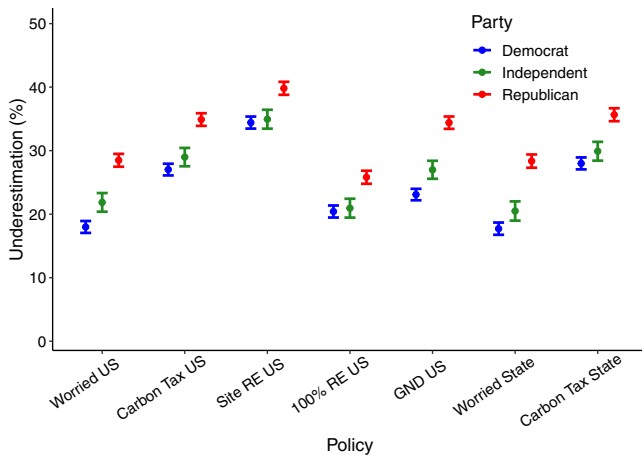

**Fig. 3 | Partisan levels of pluralistic ignorance for climate worry and support for climate policy.** Pluralistic ignorance is calculated as the difference between real and perceived norm levels, so greater values indicate real levels are higher than perceived norms (i.e. greater underestimation). All perceptions pertain to either the entire nation or one's state (not one's fellow partisans). Items on the left labeled with "US" indicate estimates of support for policies at the national level, while those on the right labeled with "State" indicate estimates of support for those in participants' own state. Error bars represent 95% confidence intervals of the mean differences illustrated. RE refers to renewable energy, and GND refers to the Green New Deal. $N = 6119$ survey participants, including 2777 Democrats, 1022 Independents, and 2320 Republicans.

states with the highest margin for Biden had pluralistic ignorance levels of 25.5%, while those with highest margin for Trump were 28.1% off. We also find a significant effect for state-level environmental protests $b = -0.47$, $t(39980) = 2.32$, $P = 0.020$, such that states with the highest level of protest were estimated to have pluralistic ignorance levels of 25.9%, while those with the fewest protests were 28.3% off.

**Variation by media consumption**

In exploratory analyses, we assessed the relationship between news media consumption and pluralistic ignorance levels across items. Using a dummy-coded variable to compare consumers of each news outlet (those who view it at least weekly) to those who do not, we assess the effect of media consumption for each outlet in a multiple regression mixed model with random intercepts for participant and item. And as media consumption may vary based on demographics, we control for the full battery of demographics assessed here, including personal political orientation, education, age, race, and income. We find that consumers of all news media outlets underestimate climate concern and policy support by around 25–30% (see Fig. 5, Panel A). Contrasting the differences between viewers and non-viewers of each outlet (see Fig. 5, Panel B), we see relatively lower levels for consumers of public broadcasting (National Public Radio), and mainstream news outlets including major national papers (e.g., New York Times), major cable news outlets (e.g., CNN), national broadcast news networks (e.g., ABC). We see relatively higher levels for those who consume news from major conservative outlets like Fox News and other conservative outlets (e.g., Breitbart), as well as for other liberal outlets (e.g., The Nation).

This pattern generally suggests that media exposure to outlets with less favorable coverage of climate change policy correspond with lower estimates of public support, with the increase in pluralistic ignorance for those consuming "other liberal outlets" as the exception. Notably, "other liberal outlets" was also the least consumed outlet, with only 15.5% regularly consuming it (95% CI = [14.6, 16.4]), while all other outlets were consumed by 20–63% of Americans. One possibility is that consumers of these liberal outlets recognize their news source

both liberal and niche, and therefore presume others do not share their more liberal, pro-climate attitudes.

## Discussion

We find that roughly 80–90% of Americans underestimate the true level of concern for climate change as well as support for transformative climate policies like a carbon tax, 100-percent renewable energy mandates, and a Green New Deal. Not only are these misperceptions nearly universal in the country, but the magnitude is large enough to fully invert the true reality of public opinion: although polls show that a supermajority support these climate policies (66–80%), the average American's estimate of public opinion suggests it is just a minority (37–3%, effect sizes of the difference ranging from $d = 0.91$–$1.48$). In other words, supporters of major climate policies outnumber opponents 2 to 1, but Americans falsely perceive nearly the opposite to be true. In fact, Americans' estimates for all national support for climate policies is roughly the same or even lower than even just Republican levels of support.

This misperception is highly robust, being present for all the climate policies assessed here, and true across the country: Americans in every state and of all major demographics are 20% or more off in their estimates of support for all climate policies. In all cases, Americans failed to understand that a strong majority of fellow Americans support climate policy, instead, estimating it to be a minority. Given both the ubiquity and magnitude of misperception, this represents a notable form of pluralistic ignorance, perhaps best described as a false social reality, defined here as a case where an inverted perception of the attitudes of others is nearly held by all in a society.

We also find preliminary evidence for possible sources of this misperception. Our results are partially consistent with previous theory and research on false consensus effects:[29] those who are less likely to support these policies (conservatives) are more likely to underestimate climate policy support by a greater degree. Our results are also consistent with previous theory and research on the conservative bias[22], where people may anchor on more conservative historic levels of political attitudes, failing to update estimates to match current public opinion. Further, consistent with availability heuristics[30], salient information from one's local norms, such as the political ideology of those in one's state, and the number of climate protests one might observe in their state, are also linked to these misperceptions, such that more liberal states and states with more climate protests have somewhat lower misperceptions. Finally, the news media that one consumes may also play a role: those who consume conservative outlets are more likely to have more erroneous views.

Beyond these contributing factors, there are additional psychological mechanisms that may help explain the effects found here. For example, it has been shown that many liberals experience "false uniqueness" whereby they falsely assume that their own opinions are less common than they really are[36], which could explain why even liberals underestimate levels of support for climate mitigation policies by a large degree. Broadly, there are many psychological factors that are plausible contributors to the misperceptions documented here and warrant future research. Additionally, media consumption may correspond with other demographic features not controlled for here and the data used in analyses here are correlational in nature. So while recent research finds that media consumption plays a causal role in shaping political beliefs[37], one should exercise caution in presuming media effect cause the pattern of results observed here until experimental data can confirm such effects.

These results have a number of concerning implications. The extent of pluralistic ignorance in this context presents at least two major hurdles for climate action. First, it undermines people's willingness to discuss the issue[8] and thus obstructs organizing around it. And second, erroneously enlarged perceptions of the opposition's numbers should increase conformity pressures to oppose climate

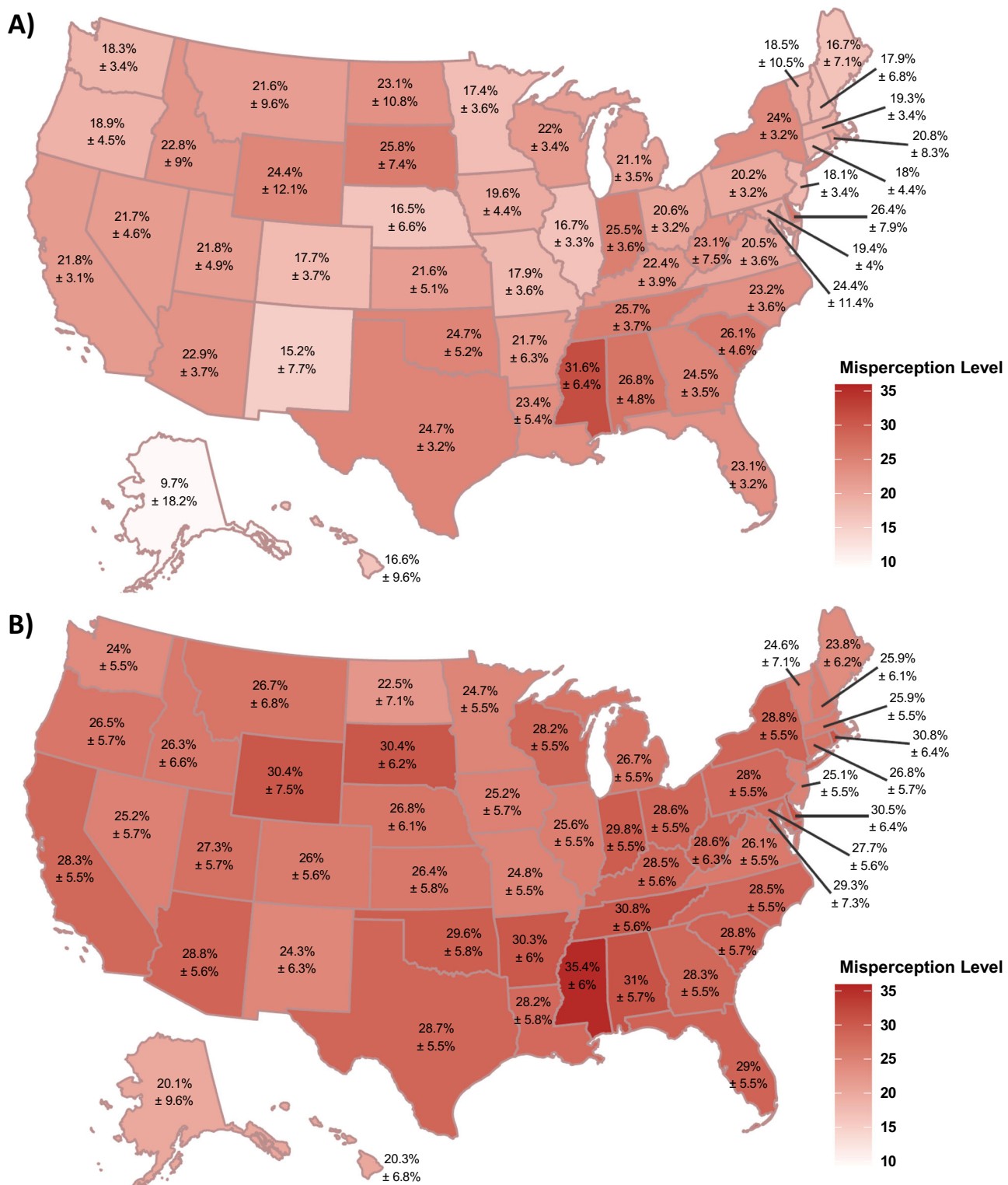

**Fig. 4 | Pluralistic ignorance for climate worry and support for climate policy across the U.S. A** shows pluralistic ignorance levels for worry about climate change by taking the difference from real and perceived national levels of climate change and averaging those levels across participants in each state. **B** shows levels of pluralistic ignorance for climate policy, calculated by averaging the difference between real and perceived support levels across the four national policies examined for each participant, and then averaging those levels of national policy support misperception for all participants in each state. $N = 6119$ survey participants. In both panels, greater values indicate real levels are higher than perceived norms (i.e. greater underestimation in perception). The ± values represent the margin of error (half of the 95% confidence interval for each state's mean).

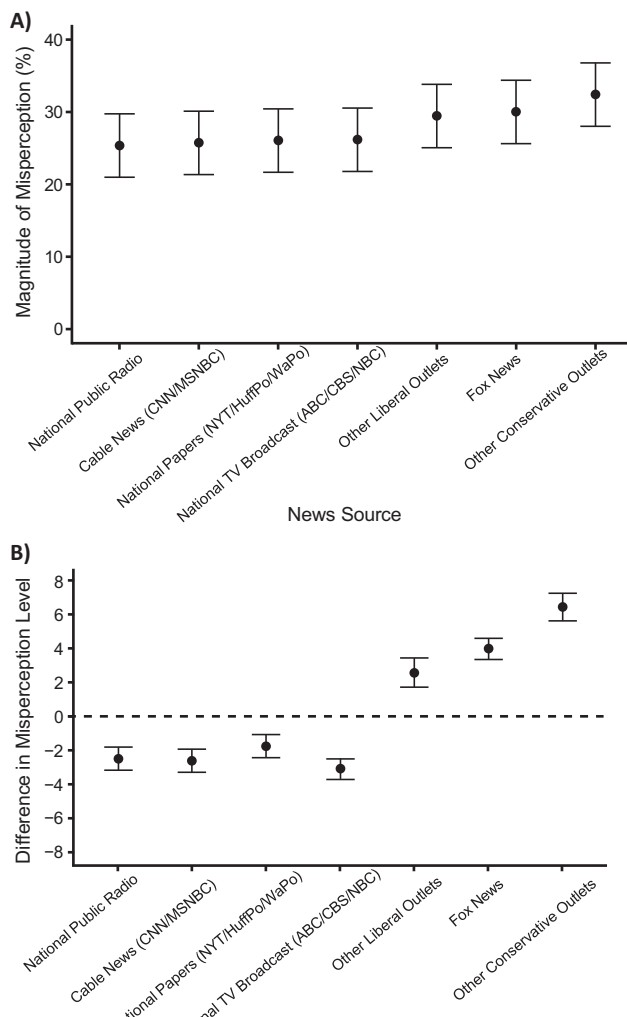

**Fig. 5 | Pluralistic ignorance and news media consumption. A** shows pluralistic ignorance levels for consumers of each news media source. **B** Shows effects on pluralistic ignorance levels comparing consumers to non-consumers of each news media source (zero = baseline levels of misestimates). Estimates in both panels are obtained from a mixed model contrasting viewers and non-viewers of each news source and controlling for participants' demographics. Pluralistic ignorance levels of $N = 6119$ participants are calculated by taking the difference from real and perceived levels of policy support and climate concern (greater values indicate real levels are higher than perceived norms, i.e. greater underestimation in perception), and controlling for the battery of demographics assessed (see Methods). Error bars are 95% confidence intervals of the means illustrated.

policy[13], diminishing motivation and political pressure to pursue these essential climate goals. Further, a perceived popular consensus around climate change may be key to reducing polarization around climate change as it can help bring conservatives closer to the majority of Americans on the issue[38]−while in the absence of this perceived consensus as seen here, polarization may thrive. If so, these misperceptions represent be a self-fulfilling prophecy: one where underappreciated levels of support for climate policy inhibit support for climate solutions needed, and undermine nascent efforts at substantive change.

The results of our study clearly establish pluralistic ignorance in the US climate policy context, and granularly maps it out, while offering an initial investigation into possible sources. Given the host of ways in which pluralistic ignorance may undermine climate policy support and action, these findings may help us understand the historic absence of major national climate policy despite solid majorities of the

American public favoring strong action and setting goals such as net zero by 2050[39]. These results also underscore the need for future research to investigate and document the variety of possible contributing factors of pluralistic ignorance, including those explored here, particularly by utilizing experimental and causal evidence. Such work may help in developing and accessing practically relevant interventions. Norm misperceptions have been addressed by interventions in a variety of domains, such as those aimed at increasing perceptions of tax compliance[40], reducing perceptions of heavy drinking on college campuses[18], and reducing perceptions of that school bullying is approved of[41]. Our work suggests the importance of developing a similar intervention in the climate policy context to correct pluralistic ignorance and help empower efforts to pass transformative climate policies.

## Methods
### Participants
We used the Ipsos eNation Omnibus nationally representative panel to survey US adults ($N = 6,119$) between April and May in 2021. Ipsos calibrates respondent characteristics to be representative of the U.S. population where source of these population targets is U.S. Census 2019 American Community Survey data, including targets for region, gender, age, and household income. To recruit a greater number of participants from less populated states, Ipsos implemented a cap on the number of participants from larger states (ending recruitment from states after they reached $N = 250$). This method allowed Ipsos to continue using their representative panel, but kept recruitment open for participants in smaller states. While the aim was not to recruit equal numbers from every state, this approach did improve recruitment of participants from small states that otherwise would have very small numbers. Ipsos provided weighed values used in all calculations of national levels to ensure representativeness. These post-hoc weights were made to the population characteristics on gender, age, race/ethnicity, region, and education. Ipsos implemented the following data quality control checks: removal of participants who took less than half of the median time, those who streaked responses in survey responses, and those who did not complete the survey. This survey provider was chosen for its high-quality data collection and for being the same provider used in polling actual levels of concern and support for identical items by the YPCCC, which is one of the most comprehensive polling efforts done on US climate opinion[11].

### Materials and procedure
For all norm estimates, participants responded using a free response question. Participants were asked to estimate the percent of Americans who were at least somewhat concerned about climate change, as well as the percent of Americans who supported each of the following climate policies: a carbon tax, a 100-percent renewable energy mandate for electricity, siting renewables on public lands, and a Green New Deal (GND). Each policy was shown given the same brief description as used in polling by the YPCCC (see Table 2). However, our phrasing did differ for the item about worry: while we asked about worry in "climate change", YPCCC asked about worry in "global warming". Another nationally representative sample of Americans[42] polled at the same time (early 2021) asked "How concerned are you about global climate change?", and found similar (slightly greater) levels of concern about (72% at least "somewhat concerned") compared to YPCCC's data on "global warming" (66% at least "somewhat worried"). Differences in concern about climate change and global warming may lead to different precise levels of pluralistic ignorance. Participants were then asked to estimate concern and support for the same policies among those in their state of residence.

In comparing perceived levels to real levels for the items in Table 1, we use a one sample t-test against a constant. Comparing against a constant value reflects the confidence held in the overall

**Table 2 | Norm items: what percent of Americans hold the following opinions?**

| Item | Wording |
|------|---------|
| Climate change Worry | Feel at least "somewhat" worried about climate change. |
| Carbon tax | Support requiring fossil fuel companies to pay a carbon tax and use the money to reduce other taxes (such as income tax) by an equal amount. |
| 100-percent Renewable energy mandate | Requiring electric utilities to produce 100% of their electricity from renewable energy sources by the year 2035. |
| Siting renewable Energy on public lands | Support generating renewable energy (solar and wind) on public land in the USA |
| Green new deal | Support a "Green New Deal" to produce jobs and strengthen America's economy by accelerating the transition from fossil fuels to clean, renewable energy. The "Deal" would generate 100% of the nation's electricity from clean, renewable sources within the next 10 years, upgrade the nation's energy grid, buildings and transportation infrastructure, increase energy efficiency, invest in "green" technology research and development, and provide training for jobs in the new "green" economy. |

*Note*. Wording used in the four policy items is the same as used in polling by YPCCC[11,12].

body of work YPCCC has collected, sampling tens of thousands of observations from nationally representative polls regularly for over a decade. However, one could choose to ignore the broader body of work and use a two-sample *t*-test comparing the perceived value to the specific poll selected for comparison. Doing so does not meaningfully change the results (Supplementary Table 1).

Next participants were asked for their media consumption of the following outlets: "Mainstream cable news (CNN, MSNBC)", "New York Times, Huffington Post, or the Washington Post", "News from ABC, CBS, NBC, or similar local or national TV broadcast news", "NPR (radio or online)", "Fox Cable News", "Other conservative news, shows or radio (Breitbart, Drudge Report, Newsmax, Rush Limbaugh, The Blaze, OAN, etc.)", and "Liberal news outlets (Democracy Now, The Intercept, The Nation, Salon, Mother Jones, Common Dreams, etc.)". These were shown in a matrix with the following frequency options: "Never", "Every few weeks", "Once a week", "Every few days", "Almost every day or more".

We used up-to-date polling data available from YPCCC, including the polling results published in 2021 on worry about climate change[43] and climate policy items[12].

For the state levels of environmental and climate protests, we utilized the protest event data collected by the Crowd Counting Consortium[44]. Of these records, we selected protests from the past 5 years published in May 2021, and pertaining to climate, the environment, and/or energy. This dataset includes protests events with as few as 1 people in attendance, and many of these smaller events lack clear confirmation. Therefore, we only included protests with at least 100 attendees. This yielded 1046 protests of interest spread across the US. We then calculated protests per capita using the 2019 US Census estimates for each states' population and logged the result to obtain a fairly normal distribution (skew = −0.28, kurtosis = 1.2; vs. skew = 6, kurtosis = 58.1 when not logged).

Demographic variables provided by IPSOS include political orientation, age, race, gender, education, income, employment, marriage and housing status, number of children, and whether participants live in an urban, rural or suburban area. This research was approved by an Internal Review Board at the home institution of the corresponding author and informed consent was obtained from all participants. Participants provided their informed consent prior to completing the survey. Data were analyzed in R (version 3.6.1)[45].

## Reporting summary
Further information on research design is available in the Nature Research Reporting Summary linked to this article.

## Data availability
All data analyzed are included in the supplementary data files (Supplementary Data File 1 for participant data; Supplementary Data File 2 for a participant data codebook; Supplementary Data File 3 for state-level data used; and Source Data for all figure source data). Source data are provided with this paper.

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

## Acknowledgements
We thank Climate Central for their support and feedback on this research. The preparation of this manuscript was supported by National Science Foundation grant, SES-DRMS 2018063 to E.U.W. and G.S.

## Author contributions
G.S., N.G. and E.U.W. designed the research, G.S. and N.G. collected the data, G.S. and N.G. analyzed the data, and G.S., N.G., and E.U.W. wrote the paper.

## Competing interests
The authors declare no competing interests.
