## [Peer Review File · Nature Communications]

Reviewer comments, initial review

Reviewer #1 (Remarks to the Author):

I enjoyed reading this piece, and I'm glad to see this attention to the topic of pluralistic ignorance around climate change. The manuscript is well written and the analysis is sound. I have only a few minor suggestions to consider before publication:

Page 6, paragraph 2: I'd suggest clarifying that the data from the Yale Climate Opinion Maps are model estimates at the state-level, based on nationally representative data. This is a minor distinction but important to clarify.

Page 6, paragraph 2-3: Did the question use the term "climate change" or "global warming"? The YPCCC surveys use "global warming," but here and in Table 2 this is not made clear. I'd suggest correcting or clarifying.

Page 6, paragraph 3: typo—remove apostrophe in "Americans' "

Page 15, paragraphs 1-2: are all the differences reported here statistically significant? For example, the difference between education categories seems small enough as to not really be meaningful.

Discussion: I'd suggest citing the following piece as additional context showing that policymakers (in addition to the public) have similarly distorted perceptions around climate concern and climate policy:

Hertel-Fernandez, A., Mildenerger, M., & Stokes, L. C. (2019). Legislative Staff and Representation in Congress. *American Political Science Review*, 113(1), 1–18.
<https://doi.org/10.1017/S0003055418000606>

Signed,
Peter Howe, Dept. of Environment and Society, Utah State University

Reviewer #2 (Remarks to the Author):

Review of A FALSE SOCIAL REALITY IN AMERICAN CLIMATE POLICY

Summary

This paper assesses the relationship between people's perceived support and actual support for four environmental policies (i.e., carbon tax, Siting RE, 100% RE, and GND), and how much people perceive others to worry about climate change and their actual worries. These relationships are tested by comparing two large-scale panel data sets from the U.S. In line with the psychological phenomena described as "pluralistic ignorance" or "norm misperception", the authors report clear and robust effects showing that people generally underestimate other people's support for climate policies and other people's worries. In further assessing the pluralistic ignorance effect, the authors assess the potential moderating effects of political affiliation, regional variation, and media consumption.

In general, the authors apply an established psychological effect to a highly relevant societal issue. The possible practical implications are noteworthy. Yet, the research questions are somewhat vague, and too little attention is given to the psychological process while much attention is devoted to media consumption, which might be confounded with other variables. Taken together, I suggest less focus on "media consumption" and a clearer test of the psychological processes driving this effect, possibly by adding relevant panel data or conducting an experiment.

Title and Abstract. The term "false social reality" reads too generic. It doesn't reflect the phenomena of interest with sufficient precision. Please consider revising it.

It is formally correct that this is the first national investigation of the accuracy of perception of support for transformation policy, it should however be noted that the pluralistic ignorance effect of environmental policies has been demonstrated before, both in the offshore wind energy using a convenience sample (Sokoloski, Markowitz, & Bidwell, 2018, as noted by the authors) and in a large (n = 8000) representative sample for "the proportion of the US population trying to fight global warming" (Andre, Boneva, Chopra, & Falk, 2021).

Page 1. Please consider adding the meta-analysis (Fischer et al., 2011) on the bystander-effect, it presents some moderators that might have important implications for this manuscript.

Ref 3, 4, 5: This seems to be an argument based on theoretical evidence, please consider citing and attending to empirical papers on the influence of social norms to promote collective actions.

Page 3. Please specify the term "supermajority". To my understanding, this can either refer to 60% or 66%.

Page 4. The author asks several important questions. First, if the general and local prevalence of pluralistic ignorance for climate change policies in the U.S. I think this is a basic and valid question that can be answered by the given data.

The second question asks if the phenomena affect "many" or "one specific" policy. The authors provide no clear theoretical rationale for why a specific policy or many policies would be affected. Moreover, the authors assess a selected number of policies (i.e. the four policies: carbon tax, siting RE, 100% RE, and GND). Could this question be adequately answered with this limited set of policies?

Finally, I found the last question, focusing on the strength of pluralistic ignorance, to be too unspecific. What is meant by "just a little off" and "more egregious"?

On page 5, the authors focus on the "reasons" for misperception. Two relevant and plausible psychological processes are presented 1) the false consensus effect and 2) the availability heuristic. I cannot see that the authors try to answer any of these questions, instead, they focus on the somewhat more peripheral question on media consumption. Given the present data set, I fully understand the focus on media consumption. Yet, I think this is a clear limitation of the manuscript. At this point, the psychological mechanisms of the effects are not sufficiently attended to.

Ref 29 Please specify that this research is limited to US media.

I also have some questions about the sample. When was the IPSOS data collected, and when was the YPCCC data collected? Moreover, why were these data sets chosen? I'm not an expert in sampling public opinion, but it makes me think if the effect would hold in other data sets. Given the robustness of the effect, there are strong reasons to believe that the effect would replicate, nevertheless, it was unclear to me why these panels were chosen.

If media consumption explains the effect, wouldn't you also predict a stronger effect for policies that media attends to compared to less attended policies? Maybe this is hard to measure, but I guess some policies (e.g., carbon tax) get more media attention than GND?

The reported relationships with media consumption might also suffer from numerous potential confounders. For example, is the level of media consumption confounded by the level of ideology, or maybe the level of education? Finally, to me as a European reader, the specific media channels are unclear. Please consider describing them in more detail.

Discussion. Please consider discussing the question of correcting norm misperceptions. See for example Dillon & Lochman, 2019

There's a large body of research on "norm misperception" that might be helpful. Some useful references might be Wenzel (2005) on misperceptions of social norms about tax compliance

References

Andre, Peter; Boneva, Teodora; Chopra, Felix; Falk, Armin (2021) : Fighting Climate Change: The Role of Norms, Preferences, and Moral Values, IZA Discussion Papers,

No. 14518, Institute of Labor Economics (IZA), Bonn

Dillon, C. E., & Lochman, J. E., (2019). Correcting for norm misperception of anti-bullying attitudes. *International Journal of Behavioral development*, 1-10.

Fischer, P., & Kainbacher, M. (2011). The bystander-effect: A meta-analytic review on bystander interventions in dangerous and non-dangerous emergencies. *Psychological Bulletin*, 137, 4, 517-537.

Sokol, R., Markowitz, E. M., & Bidwell, D. (2018). Public estimates of support for offshore wind energy: False consensus, pluralistic ignorance, and partisan effects. *Energy Policy*, 112, 45-55.

Wenzel, M. (2005). Misperceptions of social norms about tax compliance: From theory to intervention. *Journal of Economic Psychology*, 26, 6.

Reviewer #3 (Remarks to the Author):

This work demonstrates that all segments of the American public dramatically underestimate concern for climate change as well as support for climate policy. To explain this effect, the authors offer preliminary support for the false consensus effect, the conservative bias, local norms, and the impact of biased news media consumption.

I've looked for things to criticize, but the truth is I really like this paper. It is well written and well argued. The data set that this research is based on is of high quality. Its size allows for a granular level of analysis that is often lacking. The conclusions the authors draw are well supported by the data.

This paper's contribution is practical, rather than theoretical. The practical implications are enormous, however, and relevant to anyone in the business of communicating about climate change. Thus I see it as an excellent fit for *Nature Communications*. The authors use fairly dramatic language in their manuscript, but I feel that the strength of their language is fully appropriate, given the short time frame we have to act to prevent climate collapse and the magnitude of the effects they document. An enormous and robust literature demonstrates the powerful impact of social norm perception on behavior. Dispelling misconceptions about climate change concern and policy support has the potential to be a powerful lever of change.

This paper also has the potential to spur more theory-driven research aimed at rectifying the false social reality it documents. In short, I would like to see this research published.

REVIEWER COMMENTS

Reviewer #1 (Remarks to the Author):

R1.1) I enjoyed reading this piece, and I'm glad to see this attention to the topic of pluralistic ignorance around climate change. The manuscript is well written and the analysis is sound. I have only a few minor suggestions to consider before publication:

Thank you.

R1.2) Page 6, paragraph 2: I'd suggest clarifying that the data from the Yale Climate Opinion Maps are model estimates at the state-level, based on nationally representative data. This is a minor distinction but important to clarify.

We now include this point in describing the Yale's Climate Opinion Maps state-level data.

R1.3) Page 6, paragraph 2-3: Did the question use the term "climate change" or "global warming"? The YPCCC surveys use "global warming," but here and in Table 2 this is not made clear. I'd suggest correcting or clarifying.

Thank you for raising this point. We use the same questions wording for the policy items in the survey. But when we asked about worry, we asked "What percent of Americans are at least "somewhat worried" about climate change?" (as shown in Table 2), whereas the YPCCC asks "How worried are you about global warming?". This raises a possible discrepancy for the worry item. Fortunately, another nationally representative sample of Americans polled in 2021 asked "How concerned are you about global climate change?", and found similar values (if anything, slightly higher): roughly 72% of Americans were at least somewhat concerned about climate change (Johnston & McCoy, 2021). Taking these points into account, we now clarify in the manuscript (including Table 2) that just the four policy items use the exact same phrasing as YPCCC. Then we note in the Methods that the worry item we asked about pertained to 'climate change' while the YPCCC poll asked about "global warming". And then we note that contemporaneous polling suggests Americans are worried about climate change at roughly similar levels, citing the aforementioned national poll.

R1.4) Page 6, paragraph 3: typo—remove apostrophe in "Americans' "

Thank you for flagging this. It has been corrected.

R1.5) Page 15, paragraphs 1-2: are all the differences reported here statistically significant? For example, the difference between education categories seems small enough as to not really be meaningful.

All the demographic variables described under "Variation by Demographics" are statistically significant in the regression described in Table S2 (including the very small difference in

education levels). We now clarify this in text as well. But, to your point, we also emphasize that some of these differences, while statistically significant, are quite small in absolute magnitude.

R1.6) Discussion: I'd suggest citing the following piece as additional context showing that policymakers (in addition to the public) have similarly distorted perceptions around climate concern and climate policy:

Hertel-Fernandez, A., Mildenerger, M., & Stokes, L. C. (2019). Legislative Staff and Representation in Congress. *American Political Science Review*, 113(1), 1–18.

<https://doi.org/10.1017/S0003055418000606>

Thank you for highlighting this relevant work. We now tie this into our discussion.

Reviewer #2 (Remarks to the Author):

R2.1) Summary

This paper assesses the relationship between people's perceived support and actual support for four environmental policies (i.e., carbon tax, Siting RE, 100% RE, and GND), and how much people perceive others to worry about climate change and their actual worries. These relationships are tested by comparing two large-scale panel data sets from the U.S. In line with the psychological phenomena described as "pluralistic ignorance" or "norm misperception", the authors report clear and robust effects showing that people generally underestimate other people's support for climate policies and other people's worries. In further assessing the pluralistic ignorance effect, the authors assess the potential moderating effects of political affiliation, regional variation, and media consumption.

In general, the authors apply an established psychological effect to a highly relevant societal issue. The possible practical implications are noteworthy. Yet, the research questions are somewhat vague, and too little attention is given to the psychological process while much attention is devoted to media consumption, which might be confounded with other variables. Taken together, I suggest less focus on "media consumption" and a clearer test of the psychological processes driving this effect, possibly by adding relevant panel data or conducting an experiment.

Thank you. We agree the practical implications of highlighting norm misperceptions in this domain are important for this piece and, as R1 and R3 note, they are the central aim and contribution of this work. To address the concerns raised here, we have revised the piece to further hone the research question, give greater attention to the psychological processes, and discuss possible confounds with media consumption as you suggest—all of which we describe in greater detail below.

R2.2) Title and Abstract. The term “false social reality” reads too generic. It doesn’t reflect the phenomena of interest with sufficient precision. Please consider revising it.

Thank you for raising this point. We agree that the use of “false social reality”, and “social reality” in general was treated too generically and, as such, was vague in its meaning and therefore unhelpful. But, in going back through the literature to find apt terminology, we repeatedly encountered the phrase “social reality”. Theory and research from a variety of disciplines use the phrase “social reality” to describe how our perceptions, both about the world and about others, are shaped by society. These are wide ranging: they include philosophers like John Searle who, in his book "The Construction of Social Reality", describes how social reality exists as a socially agreed upon set of truths about societal phenomena (1995). Psychologists, too, use the term, such as Snyder & Swann’s research “...From Social Perception to Social Reality” describe how social perception (whether accurate or not) influences and structures social interactions (1978). And going as far back as sociologists like Émile Durkheim's and his work on “Social Realism”, such terms articulate that there are truths held at a societal level that are influential and can be observed in research (1895).

Like these classic works, we take “social reality” to be a meaningful and apt term to describe how social perceptions, especially when widely shared, play a major role in constructing one’s sense of reality pertaining to social phenomena (such as national policy support) and beyond. Using this theoretical foundation and in keeping with this usage, we feel that a “false social reality” is an apt name for a phenomenon where people, en masse, misperceive the attitudes or beliefs of those in society (i.e. as a claim about a social reality, these social perceptions would be false when evaluated). To make this a clearer and more useful term for readers, we now succinctly unpack the term “social reality” in the introduction and more clearly define “false social reality” in the discussion, using the literature and explanation provided above, in order to clarify precisely what it means as used here.

R2.3) It is formally correct that this is the first national investigation of the accuracy of perception of support for transformation policy, it should however be noted that the pluralistic ignorance effect of environmental policies has been demonstrated before, both in the offshore wind energy using a convenience sample (Sokoloski, Markowitz, & Bidwell, 2018, as noted by the authors) and in a large (n = 8000) representative sample for “the proportion of the US population trying to fight global warming” (Andre, Boneva, Chopra, & Falk, 2021).

Per editor feedback, the journal policy is to remove claims of primacy generally, which we have now done. We have also added the recent citation you note given its relevance, thank you for suggesting it.

R2.4) Page 1. Please consider adding the meta-analysis (Fischer et al., 2011) on the bystander-effect, it presents some moderators that might have important implications for this manuscript.

Thank you for this suggestion, we now cite this work and related implications.

R2.5) Ref 3, 4, 5: This seems to be an argument based on theoretical evidence, please consider citing and attending to empirical papers on the influence of social norms to promote collective actions.

As suggested, we now have added two empirical papers that demonstrate how social norms promote collective actions (Bolsen, Leeper, & Shapiro, 2014; Howe, Carr, & Walton, 2021), both of which do so in environmental domains.

R2.6) Page 3. Please specify the term “supermajority”. To my understanding, this can either refer to 60% or 66%.

We now clarify that we mean 66%.

R2.7) Page 4. The author asks several important questions. First, if the general and local prevalence of pluralistic ignorance for climate change policies in the U.S. I think this is a basic and valid question that can be answered by the given data.

The second question asks if the phenomena affect “many” or “one specific” policy. The authors provide no clear theoretical rationale for why a specific policy or many policies would be affected. Moreover, the authors assess a selected number of policies (i.e. the four policies: carbon tax, siting RE, 100% RE, and GND). Could this question be adequately answered with this limited set of policies?

We agree that we can be more precise with these research questions, and better ground them in prior research. Specifically, as to why multiple climate mitigation policies may be impacted, we now explicate our reasoning and provide a theoretical explanation based on the research cited: prior research shows belief in climate change is underestimated, which may undermine the accuracy of perceptions of popular support to address that problem (i.e. one may reason that people will not support solutions to problems they do not believe exist).

We also clarify what we previously meant by “many”: we mean a set of contemporary policies being considered by political bodies that vary in core operational characteristics such as those utilizing market instruments as opposed to mandates and more regulatory approaches, or those that facilitate investment and the creation of infrastructure. We now clarify this in the manuscript and illustrate how the policies chosen span this variety.

R2.8) Finally, I found the last question, focusing on the strength of pluralistic ignorance, to be too unspecific. What is meant by “just a little off” and “more egregious”?

We agree this terminology is too imprecise. We now clarify that our research question is about the magnitude of misperception—and in particular whether the norm misperception levels maintain accurate perceptions of the majority opinion, or whether they surpass this level to lead people to misperceive what the majority of Americans support.

R2.9) On page 5, the authors focus on the “reasons” for misperception. Two relevant and plausible psychological processes are presented 1) the false consensus effect and 2) the availability heuristic. I cannot see that the authors try to answer any of these questions, instead, they focus on the somewhat more peripheral question on media consumption. Given the present data set, I fully understand the focus on media consumption. Yet, I think this is a clear limitation of the manuscript. At this point, the psychological mechanisms of the effects are not sufficiently attended to.

We agree that the psychological processes that may drive these misperceptions are a rich and exciting avenue for research. In fact, we have developed a major grant centering on the variety of possible psychological contributors to investigate. They include false consensus and availability heuristics, as well as anchoring effects (anchoring on past public opinion levels specifically), false uniqueness (among liberals), third person effects, illusions of universality and representativeness heuristics (of conservatives), correspondence bias (of others climate inaction), and more. All of these are plausible contributors and deserve time and attention.

We are glad to say that this grant is slated to be awarded, and we plan on conducting that research over the next 3 years. As one would suspect, this goes well beyond the scope of this manuscript (depending on the findings of this planned work, we expect that the complexity of these processes will be fully unpacked over the course of several manuscripts). Thus, as Reviewer 3 underscores, the present research ultimately serves as an important foundation for research on psychological processes to be investigated, even though the present work does not aim to do so.

That said, we completely agree that the possible psychological contributors should not be overshadowed by the correlational data on media consumption. We now give greater emphasis to possible psychological contributors in the present work in three ways: First, we now order our introduction of possible contributing factors to discuss false consensus effects first, availability heuristics second, and then note media consumption last. We also clarify in the introduction which assessments were used as a preliminary investigation to see if results are consistent with false consensus effects (increased pluralistic ignorance for conservatives, who on average support climate mitigation policies less) and availability heuristics (increased pluralistic ignorance for those in conservative states and states with lower visible support for climate policies via fewer climate change protests). Second, we continue this thread in the results by highlighting which results are consistent with false consensus effects and availability heuristics, and we now elevate this content to now appear before the discussion of media consumption. And third, we recap the plausible contribution of these psychological mechanisms in the discussion and explain the need for future research on the broader variety of psychological effects that may be implicated in the findings presented.

R2.10) I also have some questions about the sample. When was the IPSOS data collected, and when was the YPCCC data collected? Moreover, why were these data sets chosen? I'm not an expert in sampling public opinion, but it makes me think if the effect

would hold in other data sets. Given the robustness of the effect, there are strong reasons to believe that the effect would replicate, nevertheless, it was unclear to me why these panels were chosen.

These are important details. Beyond describing this in the Methods, we now also relate these details in the main body of the manuscript so readers can more quickly ascertain that data sets were collected around the same time (between December 2020 and May 2021), and that the Ipsos was chosen to collect data for this study as they were the same surveyor used in the YPCCC's data (to account for any surveyor-specific data practices) and as Ipsos provides a representative sample of the US.

R2.11) Ref 29 Please specify that this research is limited to US media.

If media consumption explains the effect, wouldn't you also predict a stronger effect for policies that media attends to compared to less attended policies? Maybe this is hard to measure, but I guess some policies (e.g., carbon tax) get more media attention than GND?

We now specify that the media examined is limited to US media. Further, we agree that it is an interesting question as to whether the specific number of media mentions results in a larger impact of media effects—or alternatively one could imagine that relatively few media mentions would be just as effective if there's a diminishing return of repeated discussion in media. However, unfortunately, we do not know of an available source for media discussion frequency for these policies at this time.

R2.12) The reported relationships with media consumption might also suffer from numerous potential confounders. For example, is the level of media consumption confounded by the level of ideology, or maybe the level of education? Finally, to me as a European reader, the specific media channels are unclear. Please consider describing them in more detail.

We agree that media consumption likely corresponds with other demographic variables that can produce spurious correlations, such as personal political attitudes, as you suggest. This is why we control for political orientation in our original analysis. However, other factors may matter as well, as you note, so we now control for the full battery of demographics provided by Ipsos (political orientation, age, race, gender, education, income, employment, marriage and housing status, number of children, and whether participants live in an urban, rural or suburban area). These results are very similar to the original results, which lends some confidence that the media consumption correlations are not simply tracking these individual differences. Nonetheless, we now directly caution readers that this is not a causal test of media consumption, and may be due to other individual differences not assessed here via spurious correlations. As noted, we also de-prioritize discussing media consumption in our discussion, and now discuss possible psychological contributors earlier on in the discussion.

We also now provide short descriptors for media outlets in the results (e.g. “mainstream news outlets”, “major conservative news outlets”, etc.) to clarify the nature of these outlets for readers.

R2.12) Discussion. Please consider discussing the question of correcting norm misperceptions. See for example Dillon & Lochman, 2019

There’s a large body of research on “norm misperception” that might be helpful. Some useful references might be Wenzel (2005) on misperceptions of social norms about tax compliance

Thank you for this suggestion, and highlighting additional relevant literature for this work. We now incorporate this work at the end of our discussion where we make a clear call for future research to develop an analogous approach to correct these norm misperceptions in the climate policy context.

References

- Andre, Peter; Boneva, Teodora; Chopra, Felix; Falk, Armin (2021) : Fighting Climate Change: The Role of Norms, Preferences, and Moral Values, IZA Discussion Papers, No. 14518, Institute of Labor Economics (IZA), Bonn**
- Dillon, C, E., & Lochman, J. E., (2019). Correcting for norm misperception of anti-bullying attitudes. International Journal of Behavioral development, 1-10.**
- Fischer, P... Kainbacher, M. (2011). The bystander-effect: A meta-analytic review on bystander interventions in dangerous and non-dangerous emergencies. Psychological Bulletin, 137, 4, 517-537,.**
- Sokol,ki, R., Markowitz, E. M., & Bidwell, D. (2018). Public estimates of support for offshore wind energy: False consensus, pluralistic ignorance, and partisan effects. Energy Policy, 112, 45-55.**
- Wenzel, M (2005). Misperceptions of social norms about tax compliance: From theory to intervention. Journal of Economic Psychology, 26, 6.**

Reviewer #3 (Remarks to the Author):

R3.1) This work demonstrates that all segments of the American public dramatically underestimate concern for climate change as well as support for climate policy. To explain this effect, the authors offer preliminary support for the false consensus effect, the conservative bias, local norms, and the impact of biased news media consumption.

I’ve looked for things to criticize, but the truth is I really like this paper. It is well written and well argued. The data set that this research is based on is of high quality. Its size allows for a granular level of analysis that is often lacking. The conclusions the authors draw are well supported by the data.

Thank you—we are very glad you appreciate our work.

R3.2) This paper's contribution is practical, rather than theoretical. The practical implications are enormous, however, and relevant to anyone in the business of communicating about climate change. Thus I see it as an excellent fit for Nature Communications. The authors use fairly dramatic language in their manuscript, but I feel that the strength of their language is fully appropriate, given the short time frame we have to act to prevent climate collapse and the magnitude of the effects they document. An enormous and robust literature demonstrates the powerful impact of social norm perception on behavior. Dispelling misconceptions about climate change concern and policy support has the potential to be a powerful lever of change.

Thank you. We agree that the main contribution is describing the scope and magnitude of pluralistic ignorance in this context, and that this is very pertinent to those working on this (and related) topics.

We have revised to use plainer language (per editor feedback) but whenever possible we kept language that adequately described the phenomenon in numeric terms to emphasize magnitude (e.g. the title, from 'Severely Underestimate' to 'Underestimate... by Almost Half'). We also are careful to not strip the work of the important implications while making these revisions.

R3.3) This paper also has the potential to spur more theory-driven research aimed at rectifying the false social reality it documents. In short, I would like to see this research published.

We agree that this work lays the foundation for more theoretical work to come—some of which we are designing now.

Reviewer comments, second round review –

Reviewer #1 (Remarks to the Author):

The authors have addressed all of my comments, which were minor. This version of the paper is in great shape and I'm happy to recommend publication. It will make a nice contribution to the literature to help understand the extent and causes of American pluralistic ignorance on climate.

-Peter Howe, Utah State University

Reviewer #2 (Remarks to the Author):

I was pleased to read the revised manuscript and the authors' thorough reply to my suggestions/comments. My main concerns have now been clarified. However, I still believe that the term "false social reality" is too generic. My point was not to question if this is an established term, but rather to question if it is sufficiently specific to describe the phenomenon of interest. I leave this for the Editor to decide. For me, it's a minor issue. overall, I believe this is a strong manuscript that will be of interest across academic disciplines.

Reviewer #3 (Remarks to the Author):

The authors have done an excellent job of responding to reviewer comments (particularly Reviewer 2, who had some excellent points).

In the weeks since I first read this manuscript, I have thought of it often, and wished to share the results with colleagues and friends alike. I am excited to see it published.

REVIEWERS' COMMENTS

Reviewer #1 (Remarks to the Author):

The authors have addressed all of my comments, which were minor. This version of the paper is in great shape and I'm happy to recommend publication. It will make a nice contribution to the literature to help understand the extent and causes of American pluralistic ignorance on climate.

-Peter Howe, Utah State University

Thank you. We are glad you find the work to be in publishable form.

Reviewer #2 (Remarks to the Author):

I was pleased to read the revised manuscript and the authors' thorough reply to my suggestions/comments. My main concerns have now been clarified.

However, I still believe that the term "false social reality" is too generic. My point was not to question if this is an established term, but rather to question if it is sufficiently specific to describe the phenomenon of interest. I leave this for the Editor to decide. For me, it's a minor issue. overall, I believe this is a strong manuscript that will be of interest across academic disciplines.

We appreciate your feedback, and we are glad you see no more major issues remaining, and find the piece to be a strong one. And we appreciate the desire for clarity in terms. We believe pairing the term "false social reality" with a precise definition will clarify its meaning. We did contemplate longer terms for clarity, but they started to resemble a definition rather than a term for this phenomenon (e.g. "ubiquitous misperception of inverted norms", which is still a bit unclear).

Reviewer #3 (Remarks to the Author):

The authors have done an excellent job of responding to reviewer comments (particularly Reviewer 2, who had some excellent points).

In the weeks since I first read this manuscript, I have thought of it often, and wished to share the results with colleagues and friends alike. I am excited to see it published.

Thank you. We, too, are excited to share this work with readers.